# Copula Geo-Additive Modeling of Anaemia and Malnutrition among Children under Five Years in Angola, Senegal, and Malawi

**DOI:** 10.3390/ijerph19159080

**Published:** 2022-07-26

**Authors:** Chris Khulu, Shaun Ramroop, Faustin Habyarimana

**Affiliations:** School of Mathematics, Statistics and Computer Science, University of KwaZulu-Natal, Private Bag X01, Scottsville 3209, South Africa; ramroops@ukzn.ac.za (S.R.); habyarimanaf@ukzn.ac.za (F.H.)

**Keywords:** anaemia, malnutrition, spatial, probability, joint model

## Abstract

Notwithstanding the interventions implemented to address child mortality, anaemia and malnutrition remain a concern for the future of developing countries. Anaemia and malnutrition contribute a high proportion of the causes of childhood morbidity in Africa. The objective of this study is to jointly model anaemia and malnutrition using a copula geo-additive model. This study is a secondary data analysis where a Demographic and Health Survey of 2016 data from Angola, Malawi, and Senegal was used. The descriptive analysis was conducted in SPSS and the copula geo-additive model analysis was performed in R 3.63. The results showed that female children are notably associated with anaemia and a malnourished status (female estimate = 0.144, *p*-value = 0.027 for anaemia; female estimate = −0.105, *p*-value = 000 for malnutrition). The probability of each result decreased with an improvement in the mother’s level of schooling. This indicates an urgent requirement for interventions to be implemented by policymakers in order to manage children’s mortality rates. These interventions can include the introduction of educational programs for older adults, children’s dietary programs, and income generation initiatives (starting a small business, etc.). It is hoped that this paper can foster the utilization of copula methodology in this field of science with the use of cross-sectional data.

## 1. Introduction

Notwithstanding the interventions implemented to address child mortality, anaemia and malnutrition remains a concern for the future of developing countries [1]. Anaemia and malnutrition contribute a high proportion of the causes of childhood morbidity in Africa [2]. A report by the WHO (World Health Organization) further revealed that most children suffering from malnutrition reside in Africa and Asia [1]. In 2020, two out of five children in Africa were affected by stunted growth, whereas, twenty-seven percent of children in Africa were affected by wasting and twenty-seven percent of children were overweight.

Statistically, malnutrition is reported to impact children the most as approximately three million children die annually due to a lack of nutrition [2].

In Angola, anaemia and malnutrition are the foremost causes of child death. The number of children affected by malnutrition is reported to be increasing, with approximately eighty-five thousand children being severely malnourished. In the report issued by the World Vision (2020) [3], Angola is ranked as number one for countries that have the weakest commitment to fight malnutrition in children. In the current literature, a study by Humbwavali et al. (2019) [4] showed that in Angola, a collective exposure results in malnutrition. Children experiencing diarrhoea and the mother being the primary care giver were found to be among many factors associated with malnutrition [4]. A study by Fernandes et al. (2017) [5] in Angola showed that male children and the household source of water being a river or lake increases the chances of children’s exposure to malnutrition.

In addition, a study to understand the nutritional shortages in children under five in Angola by Fernandes et al. (2013) [6] found that for children ranging between 6 and 23 months old who had acute malnutrition, this may have been due to the results of maternal malnutrition and insufficient nutrition from breastfeeding. In contrast, an iron shortage was revealed to be related with children’s age, inflammation, and gender in a study conducted with children under five years in Angola [7]. It was further concluded that the association of anaemia with inflammation could suggest the incidence of nutritional immunity and must be further studied. A survey conducted in Angola between 2015 and 2016 showed that sixty-five percent of 6–59-month-old children were anaemic [8]. The prevalence was found to be higher in the age interval of 6–11 months and related with malnutrition.

Senegal is among the other African developing countries experiencing malnutrition [9]. Food insecurity in Senegal differs by place of residence. In urban areas of Senegal, 9% of houses are uncertain of food consumption, whereas in rural areas of Senegal, 21% of houses are uncertain of food consumption [10]. The prevalence of malnutrition in Senegal is lower when compared to other Western countries; however, seventeen percent of children are affected. Poor complementary feeding and hygiene practices are among many contributors to child malnutrition in Senegal [11]. The occurrence of malnutrition in children under five years in Senegal was reported to be sitting at sixty-seven percent in 2016 [8]. The report further revealed that an approximate average of eighty-two percent of children under five years were anaemic from 1990 to 2016. In Senegal, a study that considered the socio-economic factors of child malnutrition and examined how programs compensate for the enlarged risk facing children and mothers revealed that children of young mothers are at a disadvantage in their nutritional status [12]. The nutritional status of Senegalese children is very important at hospital admission [13]. The number of deaths of children who are under the eightieth percentile level of weight/height is 2.64 times more than the number of deaths of children above the eightieth percentile level of weight/height.

In a study by Diouf et al. (2021) [14] in Senegal, it was revealed that monitoring nutritional status, improving hygiene conditions, and endorsing good dietary practices among children could assist in fighting malnutrition. The study further showed that the factors associated with anaemia are the presence of diarrhoea, non-consumption of vegetables, incomplete immunization status, and non-consumption of meat.

In Malawi, poor diets, a lack of food, and infectious diseases are among the contributors to malnutrition [15]. Twenty-three percent of all child deaths in Malawi are related to malnutrition, whereas four percent of Malawian children are suffering from acute malnutrition. In Malawi, male children are more exposed to malnutrition when compared to female children [16]. The study further concluded that children residing in households that have an economically-empowered female as the household head are less likely to experience malnutrition. In another study conducted by Doctor and Nkhana-Salimu (2017) [17] in Malawi, it was revealed that children living in wealthy households are less likely to experience malnutrition when compared to children living in poor households. The study further showed that having diarrhoea and fever is associated with malnutrition. It was concluded in the study that public interventions that indorse access to better sanitation, water, and hygiene facilities remain important.

In a Malawian study to evaluate the effect of fish farming on health status by Aiga et al. (2009) [18], it was revealed that a lower occurrence of malnutrition was found in children residing in fish-farming households. It was also revealed in the same study that the recurrence of oil and fat intake other than never/rarely and breastfeeding practices for a recommended duration reduced children’s chances of exposure to malnutrition.

Anaemia among under five-year-old Malawian children is related with a child’s age, fever, the mother’s level of education, and residing in poor household [19]. The study concluded that community factors have a lesser effect than individual factors on childhood anaemia; however, the community factor mother’s level of schooling is significant for childhood anaemia exposure. Anaemia in Malawian children is multifactorial [20]. Genetic causes of anaemia are common in children of Malawi [20].

Anaemia is a community well-being challenge that affects high-, middle-, and low-income countries and holds the potential for momentous hostile health consequences, as well as negatively affecting progress on economic and social development. Anaemia ensues from several effects, the important contributor being iron shortage. Almost 50% of anaemia outcomes are due to iron shortage, but the percentage differs among population clusters and in different areas [9,21]. It is estimated that, globally, 50% of children in the age range of zero to five years are anaemic [10,22].

Anaemia in Malawi, Senegal, and Angola was revealed to be associated with household wealth status, the mother’s level of schooling, and the age of the child, whereas, malnutrition was found to be associated with the type of residence, mother’s level of schooling, household wealth status, and child’s age [23,24].

Little is known about the main factors related with anaemia and malnutrition in developing countries. Various statistical techniques and methods have been employed to evaluate factors related with anaemia and malnutrition. Fewer studies have been conducted to jointly model malnutrition and anaemia. The studies that have been published on this topic have used different statistical approaches and, hence, were aimed at different objectives [25]. A study by Adeyemi et al. (2019) [25] jointly modelled anaemia and malnutrition using the multivariate conditional auto regressive (MCAR) analysis. The study found a positive correlation between malnutrition and anaemia; however, the study did not explore the significance of the correlation or consider other possible outcome combinations. That would have been useful to policymakers for the implementation of interventions.

Thus, the objective of this study is to jointly model anaemia and malnutrition using a copula geo-additive model. The significance of the study includes broadening the academic theory in the understanding of the factors associated with malnutrition and anaemia, which are modelled jointly, and mapping the results. This will assist the policymakers of Malawi, Senegal and Angola in drawing up strategies to manage mortality resulting from malnutrition and anaemia.

## 2. Materials and Methods

### 2.1. Data Source

This paper is a secondary data analysis where data from a Demographic and Health Survey of 2016 from Angola, Malawi, and Senegal were used. Since this is secondary data, there was no ethical approval required. The data were attained by submitting a written application to DHS Micro for consent.

The Angola data were collected between October 2015 and March 2016 to attain socio-economic, demographic, and other information. A sample of 16,109 households was selected, which were made up of 14,379 females aged between 15 and 49 and 5684 males aged between 15 and 54. Success rate for males and females was 94% and 96%, respectively.

The Malawi data had a total sample of 27,516 houses of which 26,564 of the total sample were occupied and interviewed successfully. The technique employed to calculate the sample population was conducted in two stages. The first stage was 850 SEAs (standard enumeration areas) incorporating 677 SEAs in rural areas and 173 in urban areas. The second stage, a 33 houses per rural cluster and 30 per urban cluster, was selected with equal chance selection.

The Senegal data were implemented to respond to the well-being challenges, monitoring the progress of implemented measures, public health challenges, and community challenges. A total of 4437 households was selected, which were made up of 8865 females aged 15–49 and 3527 males aged 15–59 who were successfully interviewed. A sample of 5722 children under five years were weighed and measured to ascertain their nutritional status; 5239 under five-year-old children aged between 6 and 59 months were examined for anaemia, and 5237 were tested from the exam microscopic for malaria.

The differences in sample sizes from Angola, Malawi, and Senegal do not limit the capability to interpret the multivariate analysis because the statistical power of any test is limited by small sample size. In this study, all the three sample sizes are adequate.

The Demographic and Health Survey data from Angola, Malawi, and Senegal were merged to create a pooled sample [24,26,27,28]. The created pool sample was then used for all the analyses conducted on this study.

### 2.2. Variable

#### 2.2.1. Measurement of the Dependence Variable

The dependence variables (children less than five years old malnourished and anaemia status) were obtained from the weight-for-age (WAZ) and anaemia level variables in the DHS data.

A child is classified as malnourished when the WAZ is less than −3.0 and nourished when the WAZ is greater than −3.0. In contrast, a child is classified as anaemic when the haemoglobin level of a child is less than 9.9 g per dL, and a child is classified as not anaemic when the haemoglobin of the child is greater 9.9 g per dL.

#### 2.2.2. Measurement of Independence Variables

Demographic, health, environmental, and socio-economic fundamentals of living were identified as the important factors in malnutrition and anaemia. The framework employed to select the independence variables was similar to that used by [24,29].

The community variable incorporated in the study was a type of resident (urban or rural). Household variables incorporated in the study were mother’s level of schooling (primary, secondary, or higher), household size (0–5, 6–10, 11–15, or >15), sex of household head (male or female), wealth index (poor, middle, or rich), birth interval (<24, 24–47, or >47), marital status (unmarried, divorced, married). The household wealth index is calculated based on the living standards. It calculates household’s assets such as ownership of living stock, water and sanitation facilities, and household construction material. The full method used is explained in the paper by Khulu and Ramroop [29].

Individual level variables were sex of child (male or female), child’s age in months (<12, 12–23, 24–35, 36–47, or 48–59), childbirth order (2–3, 4–5, or >5).

### 2.3. Statistical Analyses

#### 2.3.1. Model Overview

In this study we employed a Copula model to jointly model malnutrition and anaemia. Copulas models are functions that allow us to isolate the marginal distributions from the dependency structure of a given multivariate distribution [30]. This technique is very useful in the pricing of securities that depend on many underlying securities. In mathematical terms, suppose a random vector (*D*_1_, *D*_2_, …, *D_d_*) and its marginals are continuous, then the marginals’ cumulative distribution functions (CDFs) $F_{i}\left(x\right)=\mathrm{Pr}\left[D_{i}\leq x\right]$ are continuous.

Using the likelihood integral transformation to the component, the random vector
$$\left(W_{1},\;W_{2},\ldots ,W_{d}\right)=\left(F_{1}\left(D_{1}\right),\;F_{2}\left(D_{2}\right),\ldots ,F_{d}\left(D_{d}\right)\right)$$
has a marginal that is consistently distributed on the range [0, 1]. The Copula Z contains all the data on the dependence structure between the components of $\left(D_{1},\;D_{2},\ldots ,D_{d}\right)$, whereas the marginal cumulative distribution function $F_{i}$ contains all the information on the marginal distribution of $X_{i}$.

The Copula of $\left(D_{1},\;D_{2},\ldots ,D_{d}\right)$ is defined as the joint cumulative distribution function of $\left(W_{1},\;W_{2},\ldots ,W_{d}\right)$
$$\left(W_{1},\;W_{2},\ldots ,W_{d}\right)=\mathrm{Pr}\left[W_{1}\leq w_{1},\;W\leq w_{2},\ldots ,W_{d}\leq w_{d}\right].$$

One of the advantages of copula modelling is that it joins multivariate distribution functions to one dimensional margined distribution function. This type of modelling is in contrast with the Pearson’s linear correlation coefficient when predicting the dependence structures between dependent and predictor variables.

#### 2.3.2. Family of Archimedean Copulas

Copula selection and parameter estimation is an imperative stage when adopting a copula approach to study multivariate behaviour because an appropriate fitting confirms that both the dependence structure and the dependence strength are properly signified. The dependence structure is reflected by the chosen copula family, while the dependence strength is estimated by the copula parameters. Hence, selecting an appropriate copula family and the estimation of copulas parameters require special attention. The marginal distributions and copulas have a low rate of correct classification when sample size is small; the rate of correct classification increases with increasing sample size.

Archimedean copulas form a huge family of copulas with number convenient properties and they allow for many dependence structures. Let λ denote the generator function of a copula with the below properties:
1.λ(1)=02.$\lambda^{\prime }\left(t\right)\;\leq 0,\;for\;all\;t\;\epsilon \;\left(0,1\right)$3.$\lambda^{\prime \prime }\left(t\right)\;\geq 0,\;for\;all\;t\;\epsilon \;\left(0,1\right).\;$


Now let $\lambda^{\left(-1\right)}$ denote the pseudo-inverse, which is equal to the normal inverse for t ϵ [0, λ(0)] and is equal to zero for t ≥ λ(0). The Archimedean copula is defined as
$$Z\left(u,v\right)=\lambda^{\left(-1\right)}\left(\lambda \left(u\right)+\lambda \left(v\right)\right).$$

The Archimedean copulas are symmetric, that is Z(u,v)=Z(v,u), and they are associative, that is Z(Z(u,v),w)=Z(u,Z(v,w)).

The most-used Archimedean copulas are

I.Clayton copula


$$\varphi \left(t\right)=\frac{t^{-\theta }-1}{\theta },\;\theta \;\epsilon \;\left[-1,\infty \right].$$


For θ ≥0, the Clayton copula has a lower tail dependence.

II.Gumbel copula


$$\varphi \left(t\right)={\left(-\ln\left(t\right)\right)}^{\theta },\;where\;\theta \geq 1.$$


The Gumbel copula has an upper tail dependence.

III.Frank copula


$$\varphi \left(t\right)=-\ln\left(\frac{e^{-\theta t}-1}{e^{-\theta }-1}\right),\;where\;\theta \;\neq 1$$


Frank copulas display the properties of radial symmetry and do not have a tail dependence.

The simple visualisation comparing the most-used Archimedean copulas is displayed in Figure 1. It provides an easy understand of the difference among Archimedean copulas.

#### 2.3.3. Bivariate Copula Model

Suppose $Y_{k1}$ is the anaemia status of the $k{\mathrm{th}}^{}$ child and $Y_{k2}$ is the malnutrition status of the $k{\mathrm{th}}^{}$ child. Each dependent is binary where $Y_{km}=1$ if the child has anaemia or is malnourished, otherwise $Y_{km}=0,\;m=1,\;2$. The joint probability of the event $\left(Y_{k1}=1,\;Y_{m2}=1\right)$, with a set of covariates $x_{i1}$ and $x_{i2}$ is defined as
$$P(Y_{k1}=1,\;Y_{k2}=1\;|\;x_{k1},\;x_{k2})=Z\left[P(Y_{k1}=1|x_{k1}\right);\;P\left(Y_{k2}=1|x_{k2}\right);\;\theta ].$$

Z:${\left[0,1\right]}^{2\;}\to \left[0,1\right]$ is a two-place copula function and θ is an associated parameter that measures the dependence between the two random variables.

#### 2.3.4. Copula and Link Function Selection

The suitable model was identified based on the lowest Akaike information criterion (AIC) value. AIC is suggested to be a good criterion for ascertaining the best fit for copula model [30]. Furthermore, AIC has shown that selecting the model with the minimum expected information loss is comparable to choosing model $M_{i},\;i=1,2,..,k$, that has the lowest AIC value [31]. The AIC is given by
$$AIC_{i}=-2logL_{i}+2V_{i}$$
where $L_{i}$ is extracted by altering the $V_{i}$ free parameters to exploit the likelihood that the candidate model has generated the observed data. In this study, the authors used the Gumbel copula and the link function c (“logit”, “probit”). The selection from all the possible copulas and link function was based on the lowest value of the AIC.

Figure 2 compares the copula-based link functions visually. Image (a) in Figure 2 present the visual of the standardised link function and image (b) present the visual of the standardised inverse link function. For simple comparison of the copula-based link functions, different colours were adopted. Black color is for the cloglog link function, green color is for probit link function and blue color is for logit link function.

The cross-tabulation analysis was conducted in SPSS and the multivariate analysis was conducted in R 3.6.3 package *GJRM* (Generalized Joint Regression Modelling). To complete the mapping, each country’s boundaries were obtained from DHS program and were thereafter saved as shapes in QGIS 3.4 software (QGIS Geographic Information System, https://qgis.org/en/site/index.html accessed on 1 May 2022). These shapes were imported into R 3.63 software for results’ mapping.

## 3. Results

### 3.1. Sample Characteristics

Table 1 displays the results of the prevalence of anaemia and malnutrition. From the three countries, the prevalence of anaemia was 68%, while the prevalence of malnutrition was 83%. The prevalence of both anaemia and malnutrition was 25.2%. The uncorrelated Kendall’s tau between anaemia and malnutrition was found to be statistically significant at a five percent level of significance.

Table 2 displays the occurrence of anaemia, malnutrition, and, anaemia and malnutrition together with respect to the independence variables. The prevalence of anaemia or malnutrition or both were seen to be higher in children living in rural places when compared to those living in urban places. A considerably higher occurrence of both anaemia and malnutrition was seen in female children. The prevalence of anaemia or malnutrition or both were seen to increase as children grew older.

The occurrence of anaemia or malnutrition or both in children were seen to decrease as the mother’s level of education improved. No differences in the occurrence of anaemia or malnutrition or both were seen for the child’s birth order as well as their birth interval. The occurrence of anaemia or malnutrition or both were seen to be lower in the households that were not poor.

### 3.2. Copula Selection

Table 3 show the AIC results of the Copulas that were considered to jointly model the response variables. Based on the below table, the Gumbel Copula is selected to jointly model our responses (anaemia and malnourished).

After selecting the copula, the authors further selected the link function using the AIC. The results in Table 4 indicated that the c (“logit”, “probit”) function is the best suited for the final model.

### 3.3. Fixed Effects Results

The results of the fixed effects are based on the Gumbel copula and c (“logit”, “probit”) link function. The R 3.63 package *GJRM* (Generalized Joint Regression Modelling) was used to obtain the results. Factors that were found to be not significant in the first step of the model selection were excluded from the final model. Seven factors were incorporated in the last copula model
$$Y_{i1}={ꞵ}_{0}+{ꞵ}_{1}residence+{ꞵ}_{2}child^{\prime }s\;age+\cdots +{ꞵ}_{7}birth\;order\;Y_{i2}={ꞵ}_{0}+{ꞵ}_{1}residence+{ꞵ}_{2}child^{\prime }s\;age+\cdots +{ꞵ}_{7}birth\;order$$
where $Y_{i1}$ and $Y_{i2}$ are the child’s anaemic status and child’s malnutrition status, respectively. The ${ꞵ}_{0},\;{ꞵ}_{1},\ldots ,\;{ꞵ}_{7}$ are the parameter estimates. Hence, the joint copula model of the two response variables is
$$P(Y_{i1}=1,\;Y_{i2}=1\;|\;x_{i1},\;x_{i2})=C\left[P(Y_{i1}=1|x_{i1}\right);P\left(Y_{i2}=1|x_{i2}\right);\;\theta ]$$
where $x_{i1}\;and\;x_{i2}$ are the child’s independent factors.

The marginal model results of the fixed effects are displayed in Table 5. In this study, the level of significance used is 5%. Based on the results, the children’s place of living had no significant effect on malnutrition and anaemia (rural place estimate = −0.155, *p*-value = 0.104 for malnourished; rural place estimate = 0.053, *p*-value = 0.152 for anaemia). A child’s age was found to be significantly related with a child’s anaemia status at all age categories but had no significant effect on the child’s malnourished status.

Being female was found to be significantly related with both a child’s anaemia status and malnourished status (female estimate = 0.144, *p*-value = 0.027 for anaemia; female estimate = −0.105, *p*-value = 000 for malnourished). Furthermore, the probability of each result reduced with an improvement in the mother’s level of schooling. A single improvement in the household’s wealth index was significantly related with the decrease in the likelihood of exposure to malnutrition and anaemia.

### 3.4. Spatial and Non-Linear and Spatial Effect Results

Table 6 shows the significance of the spatial and non-linear effects for the anaemia and malnutrition response variables. The unstructured spatial effect and structured spatial effect had an important effect on the probability of each dependent. The child’s age had an important non-linear effect on the probability of each dependent. The non-linear effect results of child’s age on anaemia and malnourishment are displayed in Figure 3. The likelihood of anaemia among under five-year-old children increases from the age of 0 to 30 months and, thereafter, decreases, whereas, the likelihood of malnourishment decreases as the child grows.

The region/district structured spatial effect for malnutrition and anaemia is shown in Figure 4. Based on the data and maps available, the Senegal and Malawi spatial effect was conducted at the region level. The districts or regions highlighted in light maroon resemble a risk predicted effect and are therefore related with a lower probability of the event, whereas the districts or regions highlighted in dark maroon resemble a chance effect and are related with a higher probability of the event.

Looking at Figure 4 and Figure 5, the spatial effect for malnourishment shows that Angola has districts related with a risk of malnutrition as well as districts related with a chance of anaemia. The spatial variation was imperative to manage so as to avoid a reduction in the statistical power of inference in the model, which could have led to incorrect results.

### 3.5. Estimated Joint Probability of Malnourishment and Anaemia

From the copula regression model, the joint probabilities were calculated and averaged over the districts or regions. Figure 6, Figure 7, Figure 8 and Figure 9 display the joint probabilities for each combination of the malnourished and anaemia outcomes.

Looking at Figure 6, most of the districts in Angola show a significantly high chance of a child being malnourished and anaemic. Senegal and Malawi have few regions that show a high chance of both malnourishment and anaemia in children. From Figure 7, fewer districts or regions in Angola and Senegal have a high chance of having a nourished but not anaemic child, whereas in Senegal there is a low risk of observing a child who is not malnourished but anaemic (Figure 8). Thus, this indicates that in Senegal there is a high likelihood of children being malnourished when they have anaemia. Considering Figure 9, most of the districts or regions in Angola, Senegal, and Malawi have a small chance of children being malnourished but not anaemic.

## 4. Discussion

The objective of this study was to evaluate the association between anaemia and malnutrition among children under five in Angola, Senegal, and Malawi. The model that was used to analyse the data is the joint bivariate copula regression model. This type of model enables the testing of the correlation between dependent variables while controlling for the non-linear and linear effects as well as the effect of spatial variation. The advantage of copula regression over multivariable analysis is that the normality and the linearity of the dependence between the response variables is not assumed [31] and they replicate the dataset through simulating with any type of marginal distribution. The copula-based model can model the behaviour of skewed data.

The data analysis of this study was completed in R 3.6.3 software. For the mapping, each country’s boundaries were obtained from the DHS program and were thereafter saved as shapes in QGIS 3.4 software. These shapes were imported into R 3.6.3 software for results’ mapping. The joint probability mapping results showed that the association between malnutrition and anaemia differs by districts or regions. Some regions had a stronger association between the dependent variables when compared to other districts or regions. The focus of this paper was on the likelihood of a child being malnourished and anaemic; however, other possible outcome combinations were evaluated to provide more insight in an attempt to understand the relationship between malnutrition and anaemia.

Furthermore, differentiating the association at the district or region level will lead to a more direct implementation of interventions to control the response variables. In the districts or regions where there is a stronger correlation between malnutrition and anaemia, the success of the anaemia control program can be an indicator for the success of malnutrition. For the districts or regions with a higher probability of malnourished children but not anaemic, it would be rational to assume that there are other drivers of malnutrition. Thus, implementing the interventions of malnutrition in such a district or region to control anaemia would not be effective.

The north-west districts of Angola showed a higher probability of a child being malnourished and anaemic. The government need to allocate health resources and educational resources for adults in these districts in order to control malnutrition and anaemia in Angola. The regions in Senegal and Malawi showed the least amount of variation in the joint probabilities of children being malnourished and anaemic. This might be due to the programmes or intervention that have been implemented. More attention or resource distribution between these studied countries must be given to Angola as a priority.

This study’s fixed effects results are consistent with the results from other malnutrition and anaemia separate modelling studies. The results revealed that the mother’s level of schooling, the sex of the child, and the household wealth index are significantly associated with malnutrition and anaemia [24,29,31,32]. The significant difference found for the sex of the child might have resulted from female children not exclusively breastfeeding for the first six months of life and not being provided with appropriate fortified weaning diets [33].

Another factor that was revealed to be significantly related with malnutrition is the child’s birth interval. This result is consistent with the results of the study by Govender et al. (2021) [34]. Child’s age was revealed to be significantly related with anaemia; similar results were observed in a study by Semedo et al. (2014) [32].

The limitations of this study include employing the available similar variables from the three countries’ DHS data. This resulted in important variables being excluded that exist in one country’s DHS data but not on the other; variables such as iron deficiency, etc. Hence, future studies must include the study of more developing African countries or other countries with higher child mortality rates. Other non-nutritional factors including sickle cell disease, alpha thalassemia, and parasitic infections can aggravate the anaemia of malnutrition and iron deficiency and affect growth [35,36]. Thus, future studies need to incorporate these factors into the future models as a part of exploring other avenues to intervene in an attempt to reduce anaemia.

This will assist in formulating a general policy of managing mortality and how that policy can assist in reducing the child mortality rate for African countries. Furthermore, when the next phase of demographic and health survey (DHS) data is made available, it can be combined with the data used in this study to see whether there are changes in malnutrition and anaemia risk over time, and the effect of interventions.

## 5. Conclusions

This study presents another statistical method for jointly modelling malnutrition and anaemia. The analysis and the mapping of the results were completed in R 3.63 software (R Core Team (2021). R: A language and environment for statistical computing Vienna, Austria. https://www.R-project.org/). The results of the mapping showed that there is an association between malnutrition and anaemia. This implies that the policymakers of Angola, Senegal, and Malawi can control anaemia through interventions to control malnutrition. This will save countries’ resources from being used to implement interventions for both malnutrition and anaemia.

The stronger association between malnutrition and anaemia was observed in the north-west districts of Angola when compared to other districts. This suggests that for better control of malnutrition and anaemia, the government needs to target that part of the country. The regions in Senegal and Malawi have lower probabilities and these are distributed fairly across the regions.

Based on the fixed effects analysis, the mother’s level of schooling, the sex of the child, and the household wealth index are significant factors of malnutrition and anaemia. These results are consistent with other studies on child mortality [24,29,32,33,34]. This indicates there is an urgent requirement for interventions to be implemented by policymakers in order to manage child mortality rates. These interventions can include the introduction of educational programs for older adults, child dietary programs, and income generation initiatives (starting a small business, etc.). Furthermore, Angola, Malawi, and Senegal can form a focus group that will target poor households, parental education, the importance of breastfeeding, etc. All of these will assist in reducing the prevalence of malnutrition and anaemia among young children in Angola, Senegal, and Malawi.

The literature on copula modelling is still increasing; however, it is particularly applied in actuarial science, survival analysis, and finance. It is hoped that this paper can foster the utilization of copula methodology in this field of science with the use of cross-sectional data. Hence, the future direction of research from this study is to consider the use of longitudinal data.

## Figures and Tables

**Figure 1 ijerph-19-09080-f001:**
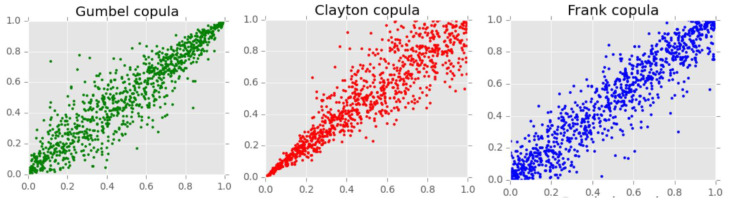
Concept visualization comparing the most-used Archimedean copulas.

**Figure 2 ijerph-19-09080-f002:**
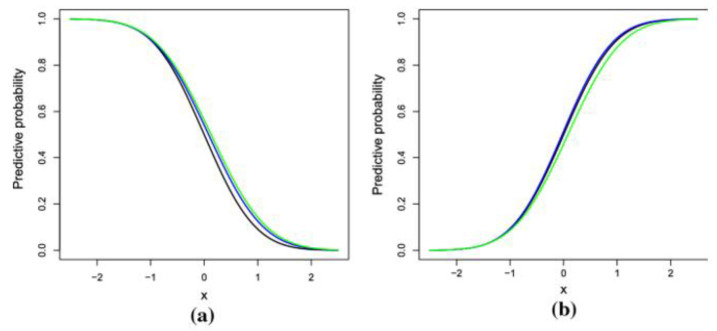
Visualization of the Copula-based link function in bivariate regression.

**Figure 3 ijerph-19-09080-f003:**
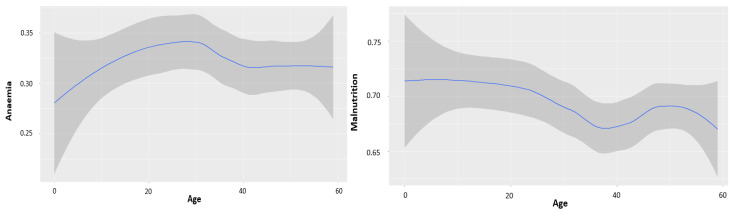
Estimated non-linear effect of the child’s age on anaemia and malnourishment.

**Figure 4 ijerph-19-09080-f004:**
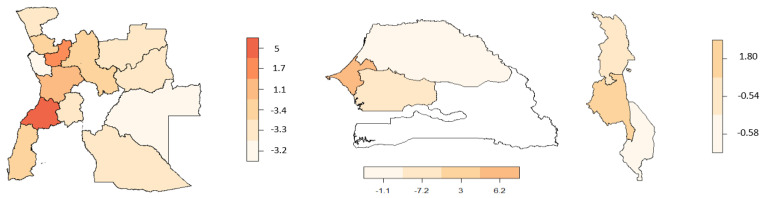
Estimated effect of the structured spatial effect on malnourishment. (**Left**) Angola, (**middle**) Senegal, and (**right**) Malawi.

**Figure 5 ijerph-19-09080-f005:**
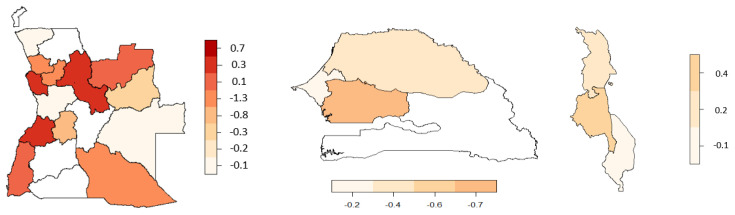
Estimated effect of the structured spatial effect on anaemia. (**Left**) Angola, (**middle**) Senegal, and (**right**) Malawi.

**Figure 6 ijerph-19-09080-f006:**
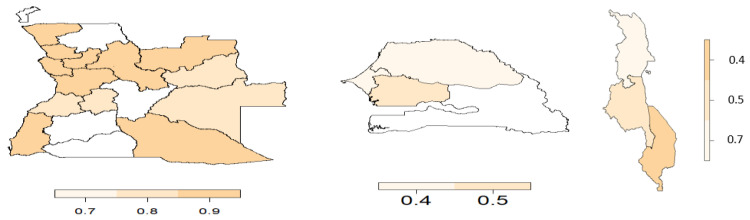
Joint probability of a child being malnourished and anaemic. (**Left**) Angola, (**middle**) Senegal, and (**right**) Malawi.

**Figure 7 ijerph-19-09080-f007:**
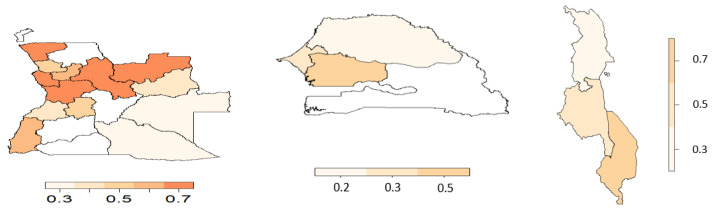
Joint probability of a child being not malnourished and not anaemic. (**Left**) Angola, (**middle**) Senegal, and (**right**) Malawi.

**Figure 8 ijerph-19-09080-f008:**
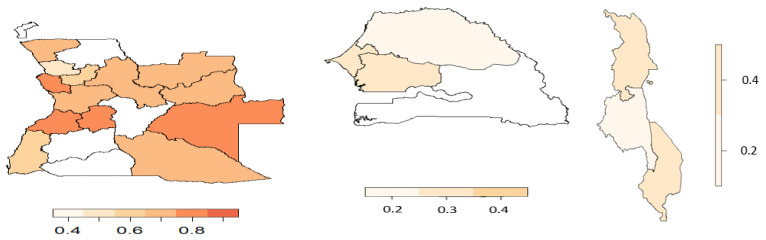
Joint probability of a child being not malnourished and anaemic. (**Left**) Angola, (**middle**) Senegal, and (**right**) Malawi.

**Figure 9 ijerph-19-09080-f009:**
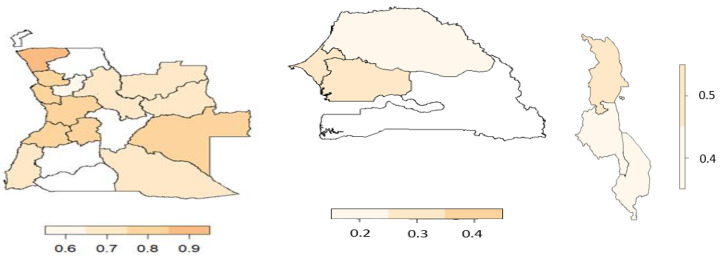
Joint probability of a child being malnourished and not anaemic. (**Left**) Angola, (**middle**) Senegal, and (**right**) Malawi.

**Table 1 ijerph-19-09080-t001:** Characteristics of the sample according to anaemia and malnutrition status.

	Malnourished	Nourished	Total
Anaemic	2916 (25.2%)	793 (6.9%)	3709 (32.1%)
Not anaemic	6677 (57.8%)	1171 (10.1%)	7848 (68%)
Total	9593 (83.0%)	1964 (17.0%)	11557

**Table 2 ijerph-19-09080-t002:** The distribution of children’s outcomes by explanatory variables.

Variables	Anaemic (%)	Malnourished (%)	Both (%)
**Type of residence**			
Rural	2133 (57.5%)	5819 (60.6%)	2514 (86.2%)
Urban	1576 (42.5%)	3738 (39.4%)	402 (13.8%)
**Child’s Sex**			
Male	1978 (53.3%)	4732 (49.3%)	1031 (35.4%)
Female	1731 (46.7%)	4861 (50.7%))	1885 (64.6%)
**Child’s Age (months)**			
<12	370 (10%)	711 (7.4%)	320 (11.0%)
12–23	819 (22.1%)	1517 (15.8%)	648 (22.2%)
24–35	860 (23.2%)	2149 (22.4%)	671 (23.0%)
36–47	931 (25.1%)	2815 (29.3%)	712 (24.4%)
48–59	729 (19.7%)	2401 (25.0%)	565 (19.4%)
**Mother level of schooling**			
Primary	1568 (42.3%)	7092 (74.0%)	2104 (72.2%)
Secondary	1608 (43.4%)	2312 (24.1%)	565 (19.4%)
Higher	533 (14.3%)	189 (1.9%)	247 (8.4%)
**Birth order number**			
2–3	824 (22.2%)	3252 (33.9%)	912 (31.3%)
4–5	1424 (38.4%)	2798 (29.2%)	934 (32.0%)
>5	1458 (39.4%)	3543 (36.9%)	1070 (36.7%)
**Birth interval (months)**			
<24	1146 (30.9%)	1516 (15.8%)	758 (26.0%)
24–47	1500 (40.4%)	4049 (42.2%)	1092 (37.4%)
>24	1063 (28.7%)	4028 (42.0%)	1066 (36.6)
**Household wealth**			
Poor	2099 (56.6%)	4141 (43.2%)	1313 (45.0%)
Middle	836 (22.5%)	2977 (31.0%)	716 (24.6%)
Not poor	774 (20.9%)	2475 (25.8%)	887 (30.4%)

**Table 3 ijerph-19-09080-t003:** The AIC results of the copula selection.

Type	df	AIC Value
Clayton	31	14,993
Gumbel	31	14,962
Frank	31	14,963

**Table 4 ijerph-19-09080-t004:** The AIC results of the link function.

Type	df	AIC Value
c (“logit”, “logit”)	31	14,947.86
c (“logit”, “cloglog”)	31	14,951.24
c (“logit”, “probit”)	31	14,947.31
c (“cloglog”, “probit”)	31	14,987.34

**Table 5 ijerph-19-09080-t005:** Parameter estimates of the fixed effects for the bivariate copula regression model of malnutrition and anaemia.

	Malnourished			Anaemic		
Covariates	Estimates	SE	*p*-Value	Estimate	SE	*p*-Value
Intercept	3.500	-	0.982	1.065	-	
Resident						
Ref: Urban						
Rural	−0.155	0.856	0.104	0.053	1.054	0.152
Child’s age						
Ref: <12						
12–23	−0.058	0.944	0.765	−0.226	0.798	0.003
24–35	0.078	1.081	0.676	−0.529	0.589	0.000
36–47	−0.037	0.964	0.840	−0.726	0.484	0.000
48–59	−0.073	0.930	0.685	−0.105	0.900	0.000
Sex of child						
Ref: Male						
Female	0.144	1.155	0.027	−0.105	0.900	0.000
Mother level						
schooling						
Ref: Higher						
Primary	1.217	2.995	0.027	1.888	6.606	0.047
Secondary	1.097	3.377	0.014	0.442	1.556	0.024
Birth interval						
Ref: <24 months						
24–47	−0.266	0.766	0.000	0.044	1.045	0.393
>47	0.265	1.303	0.020	−0.019	0.981	0.727
Wealth index						
Ref: Poor						
Middle	−0.553	0.575	0.032	0.074	1.077	0.001
Not poor	−3.444	0.032	0.000	0.226	1.254	0.000
Birth order						
Ref: 2–3						
4–5	−0.419	0.658	0.000	−0.017	0.983	0.641
>5	−0.695	0.499	0.000	−0.032	0.969	0.339

**Table 6 ijerph-19-09080-t006:** Approximate significance for the spatial and non-linear effects.

Variable	Anaemia		Malnourished	
	Chi-Square Value	*p*-Value	Chi-Square Value	*p*-Value
Childs’ age	251.01	0.003	875.09	0.011
Unstructured effect	379.32	<0.001	685.44	<0.001
Structured effect	457.77	0.045	475.33	<0.001

## Data Availability

The dataset analysed during this study is not publicly available, but data are available from the corresponding author on reasonable request. Additionally, further information about the data and conditions for access are available upon registration and request to DHS program-http://dhsprogram.com.

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
