# Peer review of "Copula Geo-Additive Modeling of Anaemia and Malnutrition among Children under Five Years in Angola, Senegal, and Malawi"

_ijerph, 2022, doi:10.3390/ijerph19159080_

Round 1

Reviewer 1 Report

Interesting. The introduction is too long and needs to be restructured. E.g. per country or per factor of importance to health.

Also use chance if positive and risk if negative. e.g risk of malnutrition

Reviewer 2 Report

Very interesting secondary analysis of anemia and nutrition in a large dataset spanning 3 countries: Demographic and Health Survey of 2016. They pull copula methodology from the field of financial analysis and demonstrate its use for multivariate analysis of public health data. 

Major critique -

1. This is a good statistical demonstration but a little deeper discussion could explore the epidemiology of the data and how future studies might use the copula methodology to answer questions about multifactorial anemia. 

2. The math modeling limitations could be discussed a little more. Are the choices of copula function and link function shown in Tables 3 and 4 important? are the analytic results sensitive to the choice of copula functions? Presumably these tools require large datasets if multiple variables are going to be modeled, such that this study pooled the data for 3 countries.  What guidelines would state the sample size needed to use these tools? If longitudinal studies are conducted, how many thousands of people need to be enrolled? 

3.  The data from Angola and Malawi appears to have intentional selection to be statistically representative with a larger number of families. The smaller data from Senegal seems to be a convenience sample. Please comment whether these differences limit the ability to interpret multivariate analysis.

Minor critique

Page 1, abstract - The abstract could emphasize one of points as the Conclusion: foster the utilization of copula methodology. Does copula add a hint that numerical analysis should use geo mapping ?

page 5, lines 208-220, the introduction of the most-used Archimedean cupola might benefit from a small diagram that compares them visually, for readers who are not accustomed to the equations.  

page 6, lines 238-240, the discussion of Gumbel copula and Logit function might benefit from a small diagram that depicts them visually. 

page 7 , tables 3 and 4 compare the AIC value to justify the selection of Gumbel copula and link function but the differences between AIC values appear to be very small.  What happens to the math model if other copula and link functions are chosen? are the models very sensitive to these choices?

page 3, lines 147- 149. Please clarify whether the "sample of 5.722 children" with is from Senegal or pooled from the three countries? If they are from Senegal, then consider merging this paragraph with the preceding paragraph (lines 143-146) so everything about Senegal is described in one paragraph.  If this is pooled from the 3 countries, then these statements in lines 147-149 should be moved to follow the statements in lines 150-152.

The gender difference in anemia is expected for menstruating females but surprising for pre-pubertal children under 5 years old. Please provide more discussion about what possible mechanisms for this difference. Are there cultural patterns of feeding girls different food than boys?

Please mention that other reasons for anemia can also have published maps of geographic distribution: sickle cell disease, alpha thalassemia, parasitic infections. These factors can aggravate the anemia of malnutrition and iron deficiency, and can also affect growth. Perhaps comment that these non-nutritional factors can be added to future models and identified as additional ways to intervene in reducing anemia. 

 Tegha G, Topazian HM, Kamthunzi P, Howard T, Tembo Z, Mvalo T, Chome N, Kumwenda W, Mkochi T, Hernandez A, Ataga KI, Hoffman IF, Ware RE. Prospective Newborn Screening for Sickle Cell Disease and Other Inherited Blood Disorders in Central Malawi. Int J Public Health. 2021 Jun 29;66:629338. doi: 10.3389/ijph.2021.629338. PMID: 34335138; PMCID: PMC8284589.

Delgadinho M, Ginete C, Santos B, Miranda A, Brito M. Genotypic Diversity among Angolan Children with Sickle Cell Anemia. Int J Environ Res Public Health. 2021 May 19;18(10):5417. doi: 10.3390/ijerph18105417. PMID: 34069401; PMCID: PMC8158763.

Minor typographic and spelling: 

page 1, line 33 "sue to lack" should be 'due to lack'

page 2, line 72 "oldness" should be 'age'

page 3 "eighty percent level of weight/height" should be 'eightieth percentile of weight/height'

page 3 "Varies statistical techniques" should be 'Various statistical techniques'

page 9, lines 316 and 318 "marron" should be 'maroon'
